# Leaf-Associated Epiphytic Fungi of *Gingko biloba*, *Pinus bungeana* and *Sabina chinensis* Exhibit Delicate Seasonal Variations

**DOI:** 10.3390/jof8060631

**Published:** 2022-06-14

**Authors:** Lijun Bao, Bo Sun, Jiayu Liu, Shiwei Zhang, Nan Xu, Xiaoran Zhang, Tsing Bohu, Zhihui Bai

**Affiliations:** 1Key Laboratory for Heavy Metal Pollution Control and Reutilization, School of Environment and Energy, Peking University Shenzhen Graduate School, Shenzhen 518055, China; baolijun@pku.edu.cn (L.B.); xunan@pkusz.edu.cn (N.X.); 2Key Laboratory of Environmental Biotechnology, Research Center for Eco-Environment Sciences, Chinese Academy of Sciences, Beijing 100085, China; bosun_st@rcees.ac.cn (B.S.); jiayu0610@foxmail.com (J.L.); zhangdan1987114@163.com (S.Z.); 3College of Resources and Environment, University of Chinese Academy of Sciences, Beijing 100049, China; 4School of Environment and Energy Engineering, Beijing University of Civil Engineering and Architecture, Beijing 102616, China; zhangxiaoran@bucea.edu.cn; 5State Key Laboratory of Lunar and Planetary Sciences, Macau University of Science and Technology, Taipa 999078, Macao; 6Xiongan Institute of Innovation, Baoding 071000, China

**Keywords:** epiphytic fungi, seasonal changes, fungal diversity, network analysis

## Abstract

Plant-leaf surface on Earth harbors complex microbial communities that influence plant productivity and health. To gain a detailed understanding of the assembly and key drivers of leaf microbial communities, especially for leaf-associated fungi, we investigated leaf-associated fungal communities in two seasons for three plant species at two sites by high-throughput sequencing. The results reveal a strong impact of growing season and plant species on fungal community composition, exhibiting clear temporal patterns in abundance and diversity. For the deciduous tree *Gingko biloba*, the number of enriched genera in May was much higher than that in October. The number of enriched genera in the two evergreen trees *Pinus bungeana* and *Sabina chinensis* was slightly higher in October than in May. Among the genus-level biomarkers, the abundances of *Alternaria*, *Cladosporium* and *Filobasidium* were significantly higher in October than in May in the three tree species. Additionally, network correlations between the leaf-associated fungi of *G. biloba* were more complex in May than those in October, containing extra negative associations, which was more obvious than the network correlation changes of leaf-associated fungi of the two evergreen plant species. Overall, the fungal diversity and community composition varied significantly between different growing seasons and host plant species.

## 1. Introduction

The above-ground surface of plants, also known as the phyllosphere, covers an estimated area of 6.4 × 10^8^ km^2^, which is approximately twice as large as the land surface area [1,2], providing a unique habitat for the colonization, interaction and evolution of microorganisms with plants. Studies have reported that there are more than 10^6^ microbial cells/cm^2^ of leaf surfaces [3]. These microbial communities include epiphytes that colonize the leaf surfaces and endophytes that occupy the interior spaces of the leaves [4,5], and have significant impacts on host plant growth, development and health [6]. There may be positive, neutral or negative interactions between microbes and plants [3,7]. Studies on phyllosphere microbiology have mainly focused on leaves, which are the plant structures with the highest global area [4]. Most leaf-associated research focuses on agriculturally relevant pathogens. Nevertheless, an increasing amount of research has demonstrated a beneficial role for the leaf microorganisms, for example, in the defense of pathogens [8] or herbivores [9]. These studies reveal new perspectives on the function of the leaf microorganisms in plant protection or growth promotion.

Plant–fungus interactions have made significant contributions to carbon and nutrients cycling, from the level of an individual plant to the entire ecosystem [10,11]. Fungal communities play significant roles in decomposition, pathogenesis and plant nutrient uptake [12,13,14], and most current research is interested in the fungal communities of the rhizosphere (the root–soil interface). On the other hand, leaf-associated fungi are abundant, and community compositions are often highly diverse [15,16] and can determine plant health and productivity. For example, fungal pathogens can cause devastating plant diseases, such as rice blast, anthracnose, leaf spot, soybean rust, wilt, blight, mildew and dieback in multiple crop species [17,18]. Plant diseases are the main cause of the reduction in the production and quality of crops [19]. In contrast, some other fungi are actually beneficial to the host plant and may promote the growth of the host plant. For example, arbuscular mycorrhiza fungi form a symbiotic relationship with host plants [20]. Non-pathogenic fungi can modify the severity of leaf pathogenesis, such as *Melampsora* leaf rust disease in *Populus trichocarpa* [21]. The rapid detection and correct identification of fungal diseases as an efficacious management tool may help to prevent and control the spread and development of fungal leaf diseases. The leaf-associated fungal communities are dynamic. Controlled microbiome management is a significant challenge, to some extent, due to the wide variety of factors that can influence the taxonomic compositions and diversities of leaf-associated fungi as well as their potential functionality. Moreover, the factors, such as host genotype, biogeography, seasonality, elevation and temperature [22,23,24,25,26], have been predicted to affect leaf-associated fungi. More work exploring the dynamics of leaf-associated fungal communities is needed to better understand the ecology and function of phyllosphere microbiota.

The overall aim of this study is to better understand the community structure and function of leaf-associated epiphytic fungi associated with urban greening plants through culture-independent methods, paving the way for improving plant vitality in the future. *Pinus bungeana*, *Sabina chinensis* and *Gingko biloba*, as the common municipal greening plants in northern China, were selected as the research objects in this study. In order to explore the seasonal dynamics of leaf epiphytic fungal communities, we determined the leaf-associated fungal communities of the three urban greenery plant species at two different sites in May and October. At the same time, differences over time were monitored to identify the main factors influencing the abundance, composition, diversity and potential function of the leaf-associated fungi in this study. Real-time quantitative PCR analysis and high-throughput sequencing methods are applied to explore the changes in the abundance and composition of leaf fungal communities in two evergreen trees, *Pinus bungeana* and *Sabina chinensis*, and the deciduous tree *Gingko biloba*.

## 2. Materials and Methods

### 2.1. Sample Collection

The sampling location, sampling protocol and sample processing was described in detail by Bao et al. [27]. Briefly, the experimental site was chosen in the city of Beijing, China, including two sampling sites, a forest park (40°0′35′′ N, 116°23′13′′ E) and a college campus (39°59′58′′ N, 116°20′6′′ E, approximately 6 km away from the forest park). At two sampling sites, leaf samples of common municipal greening plants (*Gingko biloba*, *Pinus bungeana* and *Sabina chinensis*) in northern Chinese cities were collected on 22 October 2017 and 10 May 2018, respectively. To ensure the standardization of all sample-collection conditions, we chose undamaged, healthy and mature leaves at a tree height of about 2 m. Leaf samples were collected from approximately four plants to form a composite sample for a total of five composite samples per plant species. In order to minimize the influence of other factors, all the leaf samples were collected on sunny days under similar weather conditions, and the week before sample collection was also sunny. About 300 g of composite samples collected from each plant species were placed in a Labplas on ice, and quickly transported to the laboratory. In addition, all the leaf samples from the same plant were collected during two different sampling seasons (October and May). The outside temperature of sampling was as follows: 22 October 2017 (12 °C) and 10 May 2018 (27 °C).

### 2.2. DNA Extraction

The DNA extraction method for microbiomes in leaf samples was described in our previous study [27]. Briefly, for each composite sample, 30 g of leaves were randomly selected and placed into a 1 L sterile Erlenmeyer flask containing 500 mL of sterile PBS buffer (1 × phosphate-buffered saline buffer, pH 7.4). In order to obtain the microbial cells of leaves, the Erlenmeyer flask was first sonicated (frequency 40 kHz) in an ultrasonic cleaning bath for 6 min, then shaken at 200 r/min at 30 °C for 20 min and finally sonicated (frequency 40 kHz) for 3 min. To separate the microbial cells from the leaves, we filtered the cell suspension through a 0.22 µm sterile nylon membrane. Genomic DNA on each membrane could be extracted directly by the Fast^®^ DNA SPIN Kit (MP Biomedicals, Santa Ana, CA, USA), according to the manufacturer’s instructions, after the membrane was minced with sterile scissors and stored at −80 °C.

### 2.3. Quantitative PCR Analysis

A primer set ITS1 (CTTGGTCATTTAGAGGAAGTAA)/ITS2 (GCTGCGTTCTTCATCGATGC) was used to amplify the fungal internal transcribed spacer (ITS) region with the annealing temperature of 55 °C. Quantitative PCR assay was run by a CFX96 Optical Real-Time Detection System (Bio-Rad, Laboratories, Inc. Hercules, CA, USA). The total volume of each reaction was 20 µL, which contained 0.5 µM of each primer (10 mM), 1 µL of DNA template, 10.0 µL of SYBR^®^ Premix Ex Taq (Takara, Biotech, Dalian, China) and sterile double-distilled water (ddH_2_O). To obtain a standard curve of the target fragment, an ITS region fragment obtained by PCR was cloned into the pMD19-T vector (Takara, Biotech, Dalian, China) and then transformed into Escherichia coli JM109 competent cells. Plasmid DNA in the competent cells was extracted by the Plasmid Purification Kit (Takara, Biotech, Dalian, China). We selected and verified the plasmid DNA containing the correct fragment lengths. The selected plasmid DNAs were then used as templates for generating standard curves. Sterile water was run as the template, instead of the DNA, as blank controls. Amplification of the ITS region was mainly verified by melting curve analysis and agarose gel electrophoresis. Each DNA sample was assayed in a minimum of triplicate with R^2^ values greater than 0.990 and amplification efficiencies ranging from 90% to 110%. The cycling condition was as follows: 40 cycles for 30 s at 95 °C, annealing for 30 s at 55 °C, extension at 72 °C for 30 s and a final extension at 72 °C for 8 min.

### 2.4. High-Throughput Sequencing

The extracted DNA was used as a template for the amplification of a 300-bp internal fragment of the ITS region, with a barcode primer set ITS1/ITS2. PCR analysis was performed in a reaction volume of 50 µL, containing 25 µL Premix Taq DNA polymerase, 1 µL DNA template (20 ng total DNA), 0.5 µL each forward/reverse primer (20 mM) and 23 µL ddH_2_O. Purified amplicons were pooled at equimolar concentrations and paired-end sequenced on an Illumina MiSeq PE250 platform (Majorbio, Shanghai, China), according to a standard protocol. The raw sequence data of leaf-associated fungi in October and May were deposited in the NCBI Sequence Read Archive (SRA) with the Accession Numbers PRJNA562843 and PRJNA562965, respectively. Raw reads from all fastq files were quality-filtered through Trimmomatic software (version 0.38) and barcodes were removed while assigning to the respective samples based on unique barcodes [28]. For each sample, paired-end reads were merged into full-length sequences using the FLASH software (version 1.2.11) [29]. These processed sequences were clustered by UPARSE (version 7.0.1090) [30] into operational taxonomic units (OTUs) with a sequence threshold of 97% similarity, while extracting representative sequences of the OTUs and filtering chimeras and singletons.

### 2.5. Statistical Analysis

Sequencing data were analyzed on an online platform using the Majorbio Cloud Platform at www.majorbio.com, accessed on 14 May 2019. The UNITE database (version 8.0) was used for the taxonomic identification of fungi. Based on the Bray–Curtis distance matrix, OTU-based community similarities were calculated by the vegan package of R software (version 3.3.1) using principal coordinate analysis (PCoA) and PerMANOVA analysis. Heatmap was created by R software (version 3.3.1) with vegan package and the scale represented the logarithm of the rarefied-reads number. Linear discriminant analysis (LDA) effect sizes were used to characterize the differences in fungal community compositions in different seasons, with an LDA score ≥ 2 indicating significant contributions (*p* < 0.05) to the model [31]. All the twelve-correlation network analyses were constructed at the genus level using the Maslov–Sneppen procedure [32,33], and the top 10 genera with Spearman’s correlation coefficient (*p* < 0.01 and |SpearmanCoef| > 0.75) were visualized by Cytoscape (version 3.5.1). The functional modes of the leaf-associated fungal communities were predicted by FUNGuild (version 1.0) [34]. Origin (version 2016, OriginLab, Northampton, UK) was used to create the figures. One-way analysis of variance (ANOVA) of SPSS software (version 16, IBM, New York, NY, USA) was used to calculate the significant differences (*p* < 0.05) between different groups. The statistical significance level of all the analyses in this study was 0.05.

## 3. Results

### 3.1. High Abundance and Diversity of Leaf-Associated Fungi

The abundances of fungi among the leaf samples of *G. biloba*, *P. bungeana* and *S. chinensis* from two locations over two growing seasons were detected by performing qPCR assays and were found to be different (Figure 1). The results show that ITS region copies among the leaf samples range from 2.80 × 10^6^ to 2.34 × 10^8^ copies per gram of leaf. Seasonal variation was found to be the most important factor affecting the number of ITS region copies in different leaf samples of the three plant species. The abundance of leaf samples obtained in October 2017 was significantly higher than that of the leaf samples collected in May of the following year (Figure 1A).

The fungal community structures among the leaf samples of *G. biloba*, *P. bungeana* and *S. chinensis* from two locations were assessed using high-throughput sequencing over two growing seasons. After sequences filtering and assembly, a total of 3,958,099 high-quality ITS reads were recorded from 60 detected leaf samples. Of these samples, the valid reads ranged from 41,275 to 84,327 for each sample and were rarefied to 41,275 for comparisons in leaf fungal communities in subsequent analyses (Table 1). Good’s coverage of all samples was more than 99%, indicating that the sequencing depth included nearly all fungal communities in the detected leaf samples, which was sufficient to saturate fungal diversity. The results of fungal α-diversity (Shannon and Chao1 indices) in leaf samples of *G. biloba* show significant differences between two seasons, with the two indices in May being significantly higher than those in October. However, the difference in α-diversity between October and May was not significant in the leaf samples of *P. bungeana* and *S. chinensis*, except the Chao1 index of S-C in May (Figure 1A). These results suggest that the leaf-associated fungal α-diversity of deciduous and evergreen plants may have different responses to seasonal changes. The PCoA analysis results of fungal β-diversity at the different locations and seasons reveal the obvious separation of the communities between two seasons in different samples (Figure 1B). PerMANOVA analysis further showed that changes in seasons, plant species and locations all had significant effects on the leaf-associated fungal community structures (*p* = 0.001). Moreover, seasons (R^2^ = 0.294) and tree species (R^2^ = 0.284) had a greater effect on the leaf-associated fungal communities than locations (R^2^ = 0.086).

### 3.2. Variations in Community Composition of Leaf-Associated Fungi during October and May

The rarefied dataset represented fungal communities from 8 phyla, 35 classes, 103 orders, 255 families, 531 genera and 2401 OTUs. The dominant phyla of all the detected leaf samples were Ascomycota and Basidiomycota (Figure A1). The relative abundance of the leaf-associated fungal community compositions at class level for the three plant species at two sites over two different seasons is shown in Figure 2A by performing a community bar plot analysis. Dothideomycetes was the most abundant class among all the detected leaf samples, except for S-F in May with a higher abundance of Cystobasidiomycetes than Dothideomycetes. Among these samples, S-C in October had the highest abundance of Sordariomycetes, S-F in October had the highest abundance of Eurotiomycetes, Taphrinomycetes and Ustilaginomycetes, and P-F in October had the highest abundance of Microbotryomycetes. Heatmap data displayed in Figure 2B show that the relative abundances of fungal community compositions among the three tree species over different seasons and different locations greatly vary at the genus level. The fungal community biomarkers of the three plant species in two different seasons were studied in detail by LDA analysis, and the graph shows the genus-level fungal biomarkers with LDA scores >2.5, *p* < 0.05 for each leaf sample (Figure 3). These results further demonstrate that the compositions of leaf-associated fungi at the genus level in October can be clearly distinguished from those in May among the three plant species. In detail, for deciduous tree *G. biloba*, the number of enriched fungal genera was clearly greater in May than that in October. For evergreen trees *P. bungeana* and *S. chinensis*, the number of enriched genera was higher in October than that in May, especially the samples obtained from the forest. For the three plant species, *Alternaria*, *Cladosporium* and *Filobasidium* were mostly enriched in October compared to May.

Variations in fungal communities may be closely related to changes in functions [35,36], so revealing the modes of functional transitions is important for understanding the ecological processes of leaf-associated fungal communities in different seasons or locations. The relative abundances of specific trophic modes showed that the functional modes of the same plant in different seasons or locations were different, especially the distinct shift in the functional modes of *S. chinensis* (Figure 4). It also revealed that different plant species had fungal communities with different functional modes. The results demonstrate that 93.60% of the total OTUs of leaf-associated fungi are classified into different trophic modes. Among them, saprotrophs, pathotrophs and symbiotrophs accounted for 16.25%, 6.86% and 0.51%, respectively. The remaining 69.98% of trophic modes were classified into the multiple trophic modes. For *G. biloba*, the relative abundances of fungal pathotrophs on campus decreased from 6.53% in May to 0.50% in October, and the relative abundances of fungal pathotrophs in the forest decreased from 4.85% in May to 0.81% in October. For *P. bungeana*, the relative abundances of fungal pathotrophs on campus tended to decrease from 0.71% in May to 0.22% in October, and the relative abundances of fungal pathotrophs in the forest decreased from 6.00% in May to 4.04% in October, with little difference between the different seasons. For *S. chinensis*, the relative abundances of fungal pathotrophs on campus increased from 3.05% in May to 7.12% in October, and the relative abundances of fungal pathotrophs in the forest increased from 9.15% in May to 39.31% in October.

### 3.3. Variations in Fungal Interactions of Leaf-Associated Fungi during October and May

In order to compare the differences in the interactions between the leaf-associated fungal communities of the three plant species in May and October at campus sampling sites, six correlation networks were constructed based on the genus level ITS dataset with significant (*p* < 0.01) and robust (q > 0.75 or q < −0.75) correlations (Figure 5). For *G. biloba* leaf-associated fungal communities, the correlation networks in October and May consisted of 123 pairs with 82 nodes, and 192 pairs with 123 nodes, respectively. For *P. bungeana* leaf-associated fungal communities, the correlation networks in October and May consisted of 84 pairs with 38 nodes, and 97 pairs with 53 nodes, respectively. For *S. chinensis* leaf-associated fungal communities, the correlation networks in October and May consisted of 125 pairs with 80 nodes, and 117 pairs with 85 nodes, respectively. Among the tested three plants, the changes in fungal network structures of deciduous tree *G. biloba* in different seasons were greater than those of evergreen trees *P. bungeana* and *S. chinensis*.

Six correlation networks with significant (*p* < 0.01) and robust (q > 0.75 or q < −0.75) correlations were constructed to understand the differences in the interactions of the leaf-associated fungi in different seasons on forest sampling sites (Figure 6). The leaf fungal networks of *G. biloba* consisted of 64 pairs with 47 nodes, and 180 pairs with 107 nodes in October and May, respectively. The leaf fungal networks of *P. bungeana* consisted of 98 pairs with 57 nodes, and 77 pairs with 63 nodes in October and May, respectively. The leaf fungal networks of *S. chinensis* consisted of 111 pairs with 85 nodes, and 104 pairs with 64 nodes in October and May, respectively. Among the tested three plants, the fungal network structures of *G. biloba* in May were more complicated than that in October both at two sampling sites, especially in forest sampling sites. These results indicate that seasonal factors have a great influence on the interactions between the fungal community compositions of deciduous tree, and the influence is greater than that on evergreen trees *P. bungeana* and *S. chinensis*. Moreover, the number of negative interactions between the *G. biloba* leaf-associated fungi in May were much higher than that in October. It is inferred that competitive interaction might be one of the reasons why the total abundance of fungi in May was lower than that in October.

## 4. Discussion

Diverse microorganisms, including fungi, live on plant leaves, which can affect plant growth, development and even evolution [3,6]. In spite of a growing recognition of the importance of these microorganisms to plants and even ecosystems and their potential for crop improvement, the factors that influence phyllosphere microbial communities and their functional consequences require more research to thoroughly explore them. As a step towards a more comprehensive understanding of the dynamics of leaf-associated fungal community in different plant species, we investigated the leaf-associated fungi of three plant species during two growing seasons at different locations. Our study demonstrated clear differences in leaf-associated fungal communities between different growing seasons and plant species in the northern region, whereas small geographic distance seemed to have little effect on the leaf-associated fungal communities.

Ascomycetes (Dothideomycetes, Eurotiomycetes and Sordariomycetes) and Basidiomycetes (Cystobasidiomycetes, Tremellomycetes and Microbotryomycetes) dominated the epiphytic fungal communities in the studied plant species, in line with the results of previous studies [37,38]. Leaf-associated fungal community structure varied strongly throughout the different growing seasons; the core fungal taxa, especially, showed distinct seasonal abundance patterns at higher taxonomic levels. Katsoula et al. [39] observed a strong seasonal effect on the composition of the leaf epiphytic fungal communities in a semi-arid Mediterranean ecosystem (summer vs. winter). The abundance of the total epiphytic fungal community decreased significantly from autumn to spring, which was contrary to the findings of olive trees in Mediterranean ecosystems [40]. Members of Dothideomycetes (Ascomycota) were at their highest abundance among all the detected leaf samples in the growing seasons of October and May. Dothideomycetes is the largest and most ecologically and functionally diverse class of fungi, containing human and plant pathogens, endophytes and epiphytes [41,42], which has been widely reported as a dominant leaf-associated taxa globally [38,43,44]. The relative abundance of Cystobasidiomycetes (Basidiomycota) in May was much higher than that in October, only in evergreen trees *P. bungeana* and *S. chinensis*. At the genus level, the core fungal microorganisms in this study included *Phoma*, *Cladosporium*, *Epicoccum* and *Alternaria*, which were found to be dominant in previous studies of other plant leaf-associated fungal communities [45,46,47,48]. The previous study using culture-dependent methods found that *Alternaria*, *Cladosporium*, *Fusraium* and *Phoma* were the main epiphytic fungi on Hornbeams [49], which were both included in our study with culture-independent methods. Furthermore, studies have shown that *Epicoccum* is an important endophytic fungus involved in the production of secondary metabolites, and some species, such as *E. nigrum*, can act as biocontrol agents to promote plant growth and resist brown-rot pathogen [50,51]. Therefore, the presence of this genus is of potential interest for future studies. Interestingly, the enrichment of *Filobasidium* in October might be related to a novel radiation resistance of *Filobasidium* sp. [52].

In addition to seasonal effects, host plant species have been observed to explain shifts in leaf-associated microbial communities [53,54], which follows the variations in fungal communities among different plant species in our study. The leaf-associated fungal abundance of *Ginkgo biloba* was significantly higher in May than in October, which may be related to the different maturation of leaves. Studies have shown that significant differences in epiphytic fungal richness were observed among olives obtained in different production systems and maturation stages [55]; whereas, in the same production system, there was no statistical significance in richness regarding olive cultivars, and this result is consistent with the reports of epiphytic fungal communities in olives [56] and mango fruits [57] from different cultivars studied using culture-dependent methods. The screening role of host plant species on the epiphytic fungal communities [38,43] has been fully confirmed and attributed to different ecological strategies as well as the chemical and functional properties of host plants. Different plant species have several different plant traits, including average plant height, leaf nitrogen concentration and plant genotypes. Previous studies have shown that plant height is closely related to changes in leaf-associated bacterial communities [58,59]. Studies of leaf-associated microbiomes in a variety of plant types showed that variations in the leaf-associated fungal communities were significantly related to the changes in leaf mass and nitrogen concentration per leaf area [25,37]. Genotypes play a critical effect in determining the colonization and establishment of leaf-associated microbial communities in plant species [43,60]. Many phenotypic properties derived from the plant genetic repertoire, such as leaf morphology, physiology and chemistry, may affect the interaction of plants with herbivores, pathogens, competitors and symbionts, including the leaf-associated microorganisms [61]. Leaf fungal communities may be subject to selective pressures derived from specific dynamics arising from plant phenotypes, resulting in the adaptation of local communities to individual host genotypes [62]. A previous study that sampled 32 *Arabidopsis thaliana* individuals and collected 21 air samples over a 73 day period confirmed that leaf microbial community compositions initially reflected the airborne community, but then formed a unique community, revealing great host selectively for community members [63]. A plausible mechanism for the effect of genotype on leaf-associated microbial communities may be related to leaf-surface properties, such as leaf wax [53], ethylene perception [53], gamma-aminobutyrate pathway [64], or salicylic acid and jasmonic acid signaling defense pathways [65]. Genome-wide association studies revealed that the variation in the leaf-associated microbial community was significantly associated with host plant loci responsible for defense and cell wall integrity [66]. It can be seen that the mechanisms of deciphering the associations of host plants to form microbial communities is highly complex, and more extensive and in-depth research is required.

Epiphytes inhabiting the phyllosphere are under a set of selective pressures, which are distinct from those that endophytes are facing [67]. The epiphytic community was found to be significantly richer and more abundant than the endophytic community in some woody plant systems [40,68,69]. There was a great difference in the effects of seasons on the total abundance of phyllosphere epiphytic fungi and endophytic fungi. Gomes et al. [40] found that seasons had a significant effect on the total abundance of epiphytic fungi, but had no significant effect on endophytic fungi. It is significant to compare the phyllosphere epiphytic and endophytic fungal communities in the future studies.

Functional mode analysis found that, among all the samples, only *Sabina chinensis* at the forest sampling site in October had the highest attributes of leaf fungal pathotrophs, indicating that the host plants may be infected by pathogenic fungi, which requires us to pay more attention to the disease and health of *S. chinensis*. The network among leaf-associated fungi of the deciduous tree *G. biloba* in May was more complex than that in October, showing more negative correlations, which was consistent with the previous studies on bacteria, but not with changes in the fungal network of the evergreen trees *P. bungeana* and *S. chinensis*. The unstable state of competition for space and nutrients in leaf-associated fungi of the deciduous tree *G. biloba* in May could have resulted in lower overall fungal numbers than in October. Positive correlations in fungal–fungal network correlations were more common than negative correlations in leaf-associated core taxa associations in the microbial co-occurrence networks of the evergreen trees *P. bungeana* and *S. chinensis*, in agreement with the recent reports [25]. Among the core taxa associations in the leaf-associated microbial co-occurrence networks, fungal–bacterial associations were more likely to be negative than positive [25,70,71]. Studies of *Arabidopsis* root microorganisms have shown that these negative correlations reflect direct or indirect bacterial biocontrol of detrimental fungi, which have significant implications for plant survival and health [71]. In future studies, we need to explore more comprehensive microbial network correlations, including fungal–fungal network correlations, bacterial–bacterial network correlations and fungal–bacterial network correlations, to protect plants against fungi and oomycetes through biological control methods.

## 5. Conclusions

We extended the knowledge about the leaf-associated epiphytic fungal microbiomes of three municipal greening plants, *Pinus bungeana*, *Sabina chinensis* and *Gingko biloba*, in northern cities of China. The leaf-associated epiphytic fungal communities from two different sites over two seasons were compared, revealing the strong impact of growing seasons and plant species on fungal community composition, exhibiting clear temporal patterns in abundance and diversity. Further investigations will elucidate the mechanism of this effect. Among the genus-level biomarkers, the abundances of *Alternaria*, *Cladosporium* and *Filobasidium* were significantly higher in October than in May in the three tree species. Network correlations between the leaf-associated fungi of *G. biloba* were more complex in May than in October, containing extra-negative associations, which was more obvious than the network correlation changes in leaf-associated fungi of the two evergreen plant species. Further studies will be conducted to investigate biomarkers as indicators of total microbial network correlations for protecting plant health.

## Figures and Tables

**Figure 1 jof-08-00631-f001:**
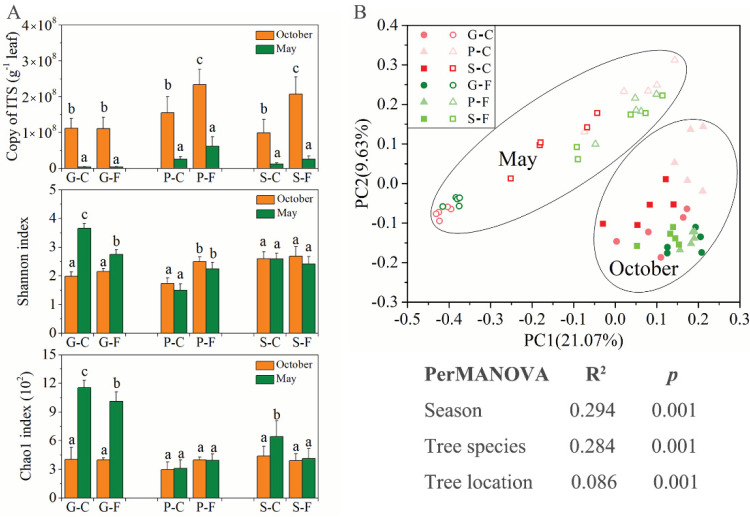
The copy number of ITS region and α-diversity (Shannon and chao1 indices) of fungal community structures of three plant types from two locations over two seasons (**A**). Principal coordinate analysis (PCoA) of the variation in fungal community structures of three plant types from two locations over two seasons (**B**). a, b, and c in (**A**) indicate significant differences between seasons and locations within a plant species. Samples (points) are colored based on different habitats and plant species. Solid points indicate samples obtained on 22 October 2017; the hollow points indicate samples obtained on 10 May 2018. G-C, *G. biloba* from campus; G-F, *G. biloba* from the forest; P-C, *P. bungeana* from campus; P-F, *P. bungeana* from the forest; S-C, *S. chinensis* from campus; S-F, *S. chinensis* from the forest.

**Figure 2 jof-08-00631-f002:**
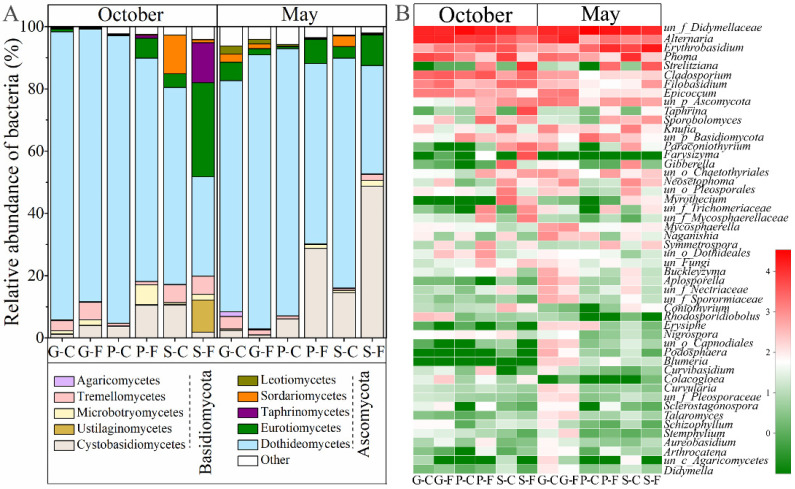
The relative abundance of leaf-associated fungi at class (**A**) and genus (**B**) level of three types of plants from two locations over two seasons. un, unclassified; G-C, *G. biloba* from campus; G-F, *G. biloba* from the forest; P-C, *P. bungeana* from campus; P-F, *P. bungeana* from the forest; S-C, *S. chinensis* from campus; S-F, *S. chinensis* from the forest.

**Figure 3 jof-08-00631-f003:**
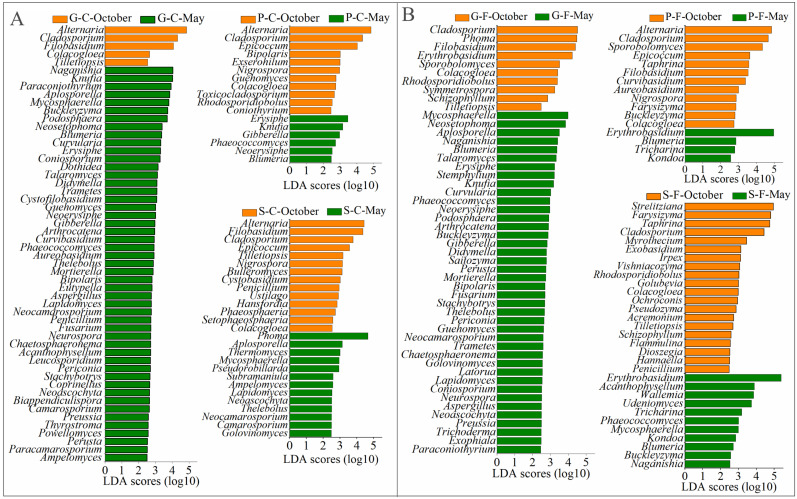
Taxonomic differences at genus level among phyllospheric fungi in October and May of three types of trees in campus (**A**) and forest (**B**) by a linear discriminant analysis (LDA). G-C-October, *G. biloba* from campus obtained in October; G-C-May, *G. biloba* from campus obtained in May; P-C-October, *P. bungeana* from campus obtained in October; P-C-May, *P. bungeana* from campus obtained in May; S-C-October, *S. chinensis* from campus obtained in October; S-C-May, *S. chinensis* from campus obtained in May; G-F-October, *G. biloba* from forest obtained in October; G-F-May, *G. biloba* from forest obtained in May; P-F-October, *P. bungeana* from forest obtained in October; P-F-May, *P. bungeana* from forest obtained in May; S-F-October, *S. chinensis* from forest obtained in October; S-F-May, *S. chinensis* from forest obtained in May.

**Figure 4 jof-08-00631-f004:**
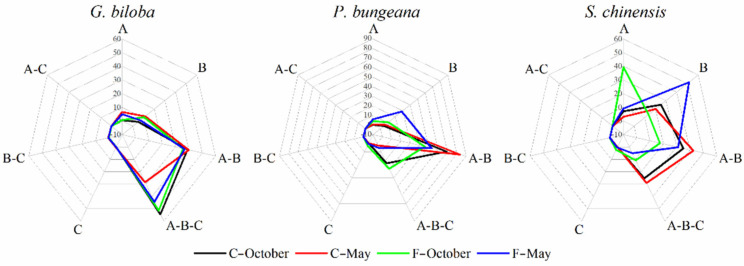
Changes in fungal metabolic function attributes of *G. biloba*, *P. bungeana* and *S. chinensis* in October and May. Letters A, B and C represent the functional attributes of pathotrophs, saprotrophs and symbiotrophs, respectively. C-October, C-May, F-October and F-May represent samples obtained from campus in October, campus in May, forest in October and forest in May.

**Figure 5 jof-08-00631-f005:**
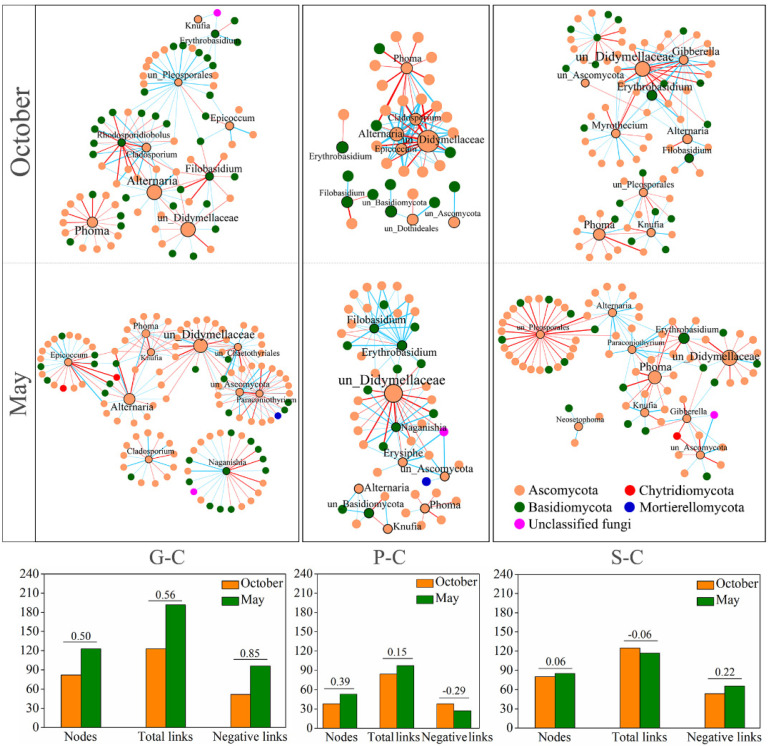
Fungal ITS region genera-based correlation network for *G. biloba*, *P. bungeana* and *S. chinensis* on campus in October and May. A node represents a genus. A connection stands for a strong (Spearman’s q > 0.75 or q < −0.75) and significant (*p* < 0.01) correlation. Edge widths are scaled according to their weights and edge colors indicate a positive (blue) or negative (red) correlation for the nodes they connect. The numbers on columns indicate the ratio of increase in May compared to October. G-C, *G. biloba* from campus; P-C, *P. bungeana* from campus; S-C, *S. chinensis* from campus.

**Figure 6 jof-08-00631-f006:**
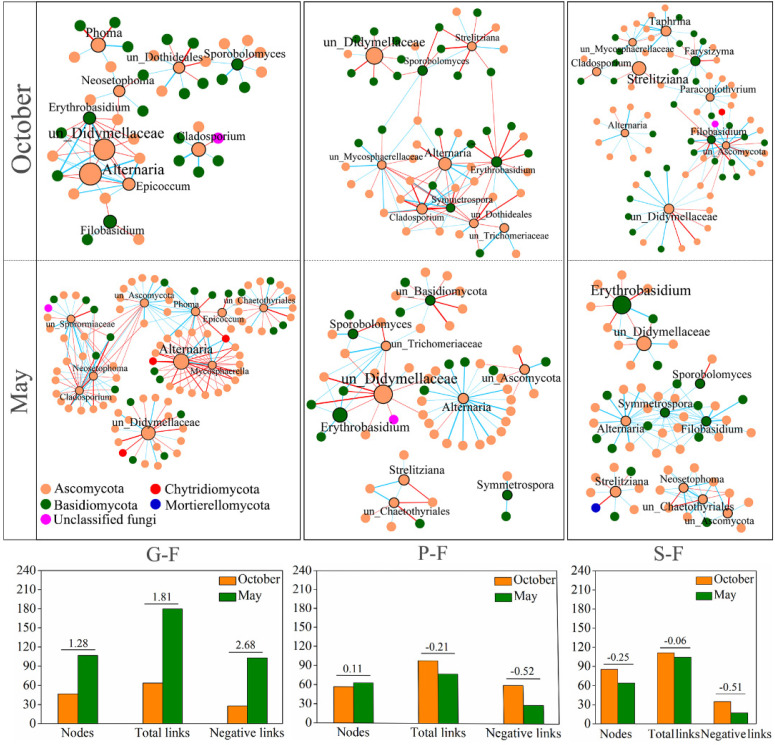
Fungal ITS region genera-based correlation network for *G. biloba*, *P. bungeana* and *S. chinensis* at forest in October and May. A node represents a genus. A connection stands for a strong (Spearman’s q > 0.75 or q < −0.75) and significant (*p* < 0.01) correlation. Edge widths are scaled according to their weights and edge colors indicate a positive (blue) or negative (red) correlation for the nodes they connect. The numbers on columns indicate the ratio of increase in May compared to October. G-F, *G. biloba* from forest; P-F, *P. bungeana* from forest; S-F, *S. chinensis* from forest.

**Table 1 jof-08-00631-t001:** Sequencing summary of each sample.

Season	Sample ID ^1^	No. of Reads	No. of OTUs	Coverage (10^−2^)
October	G-C	66,537 ± 4457	244 ± 68	99.8 ± 0.07
	G-F	66,992 ± 6242	236 ± 21	99.8 ± 0.02
	P-C	67,902 ± 7380	159 ± 34	99.8 ± 0.04
	P-F	64,663 ± 4559	275 ± 26	99.8 ± 0.02
	S-C	71,866 ± 7404	271 ± 60	99.7 ± 0.07
	S-F	58,039 ± 7050	266 ± 43	99.8 ± 0.05
May	G-C	65,209 ± 11,900	909 ± 75	99.4 ± 0.06
	G-F	68,522 ± 3565	757 ± 57	99.4 ± 0.05
	P-C	62,256 ± 8731	171 ± 57	99.8 ± 0.07
	P-F	66,604 ± 6499	234 ± 47	99.8 ± 0.05
	S-C	71,771 ± 3424	360 ± 104	99.6 ± 0.10
	S-F	65,835 ± 13,643	254 ± 79	99.8 ± 0.07

^1^ G-C, *G. biloba* from campus; G-F, *G. biloba* from the forest; P-C, *P. bungeana* from campus; P-F, *P. bungeana* from the forest; S-C, *S. chinensis* from campus; S-F, *S. chinensis* from the forest.

## Data Availability

The raw reads of leaf samples obtained in October and May in the ITS region were submitted in the Sequence Read Archive (SRA) at the National Center for biotechnology information (NCBI), with accession numbers SRP219629 and SRP219695, respectively.

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
