# Peer review of "Leaf-Associated Epiphytic Fungi of Gingko biloba, Pinus bungeana and Sabina chinensis Exhibit Delicate Seasonal Variations"

_jof, 2022, doi:10.3390/jof8060631_

Round 1

Reviewer 1 Report

Dear Author 

In response to manuscript idea is basically acceptable and appreciated for publication but I found a references that is not be relevant to idea described. It will be No. 60 in list of references which is not  applicable apparently to this scientific point. Because Ref. No. 60 is about powdery mildew? whereas is not clear why author inserted here. I wish to be clear out by the author that reference No.60 before final  correction.

With he best regard

Kind 

Reviewer 2 Report

Thank you for investigating phyllosphere fungal communities in two seasons for three plant species at two sites by high-throughput sequencing. These studies are really important to help researchers deeper understand the mechanisms underlying species coexistence and ecosystem stability. I think the article is valuable, but I have some major comments (please see below and attached PDF).

Introduction

The information in the introduction section needs to be improved. There are some questions that could help the authors to improve the introduction. What is the objective of the study? What are the previous studies related to this topic? Are there any studies comparing epiphytic fungal communities? Why this study is important? Who are epiphytic fungi? Importance of epiphytic fungi? Why the study of epiphytic fungi in the phyllosphere in different tree species/seasons is important? and what is the gap that this study is filling up?

Methodology

Phyllosphere is refer to the total above-ground surface of a plant when viewed as a habitat for microorganisms. Not only leaf surface. I reckon this study is only based on leaf surface associated fungal communities. Therefore, I recommend the authors to use leaf-associated epiphytic fungi (not whole phyllosphere).

Why did you select Gingko biloba, Pinus bungeana and Sabina chinensis

Results

Provide a table for numbers of reads, and OTUs for the epiphytic fungi associated with different plant species in different seasons

Discussion

The authors should consider adding and discussing information about:

Are these results in accordance with the findings of some previous studies using culture dependent methods?

Does plant identity significantly influence the richness of endophytic fungi?

Are there any significantly different scenarios to those are reported in previous studies?

Is there any data that you could compare with endophytic fungi?

Finally, the importance of this study for future researchers and fungal studies in these hosts

Final conclusion is missing

Round 2

Reviewer 2 Report

The authors have carefully addressed the comments of the reviewers. The revised version of the manuscript has been significantly improved. This new version can be accepted for publication.